# Epidemiological clinical profile and closure of chronic plantar ulcers in patients with leprosy sequelae undergoing orthopedic surgery in a municipality in western Amazon

**Francisco Mateus João**[1,2,3]\*, **Silmara Navarro Peninni**[4], **Zanair Soares Vasconcelos**[2,5], **Arineia Soares da Silva**[1,2], **Katia do Nascimento Couceiro**[1], **Alba Regina Jorge Brandão**[1,2], **Mônica Regina Hosannah da Silva e Silva**[1,2], **Marcello Facundo do Vale Filho**[6], **Guilherme Miranda Silva de Oliveira**[7], **Lucas Silva Ferreira**[1,2], **Victor Irungu Mwangi**[1,2], **Bernardo Maia da Silva**[8], **Maria das Graças Vale Barbosa Guerra**[1,4,9], **Jorge Augusto de Oliveira Guerra**[1,6,9]

1 University of the State of Amazonas, Manaus, Amazonas, Brazil, 2 Postgraduate Program in Tropical Medicine, Manaus, Amazonas, Brazil, 3 Adventist Hospital, Manaus, Amazonas, Brazil, 4 Alfredo da Mata Foundation Hospital, Manaus, Amazonas, Brazil, 5 Jungle Warfare Instruction Center, Manaus, Amazonas, Brazil, 6 Metropolitan Faculty of Manaus, Manaus, Amazonas, Brazil, 7 Federal University of Amazonas, Manaus, Amazonas, Brazil, 8 Carlos Borborema Clinical Research Institute, Manaus, Amazonas, Brazil, 9 Doutor Heitor Vieira Dourado Tropical Medicine Foundation, Manaus, Amazonas, Brazil

\* fmmateus@uol.com.br

**Data Availability Statement:** All relevant data are within the paper and its Supporting Information files.

## Abstract

### Introduction

Chronic plantar ulcers in leprosy are lesions resulting from motor and sensory alterations caused by Mycobacterium leprae. They are lesions refractory to conventional dressings and present high recurrence rates.

### Objective

To evaluate the epidemiological clinical profile of patients with chronic plantar ulcers associated with bony prominences in the lesion bed and to evaluate the efficacy of orthopedic surgical treatment of these lesions.

### Methods

This is a descriptive and analytical retrospective study with the evaluation of medical records of patients undergoing surgical treatment of chronic plantar ulcers from 2008 to 2018. The surgical technique applied consisted of corrective resection of bone prominences and the primary closure of the lesion with bipediculated local flap.

### Results

234 patients were submitted to surgery, 55.1% male with an average age of 69.5 years old. Of these, 82.9% were illiterate; and 88.5% with open lesions over 10 years. After surgical treatment, total wound healing occurred in an average time of 12 weeks. The variables that

**Funding:** The author(s) received no specific funding for this work.

**Competing interests:** The authors have declared that no competing interests exist.

contributed to shorter healing time were: Patients' lower age group; regular use of orthopedic shoes and insoles and dressings performed by nurse aides in health units before surgery. Obesity was the factor that correlated with the delay of healing time.

## Conclusion

A higher incidence was observed in males and male and female illiterate patients. The regular use of shoes and insoles and dressings performed by nurse aides in health units contributed to shorter postoperative healing time. Orthopedic surgical treatment with corrective resection of bony prominences proved to be an efficient therapeutic method for the closure of chronic plantar ulcers. It is a reproducible method, justifying the importance of the orthopedic surgeon in the context of the multidisciplinary team to cope with these complex lesions.

## Introduction

Leprosy is a chronic infectious disease caused by *Mycobacterium leprae*, with tropism for the skin and peripheral nervous system. It may present a slow and progressive evolution, leading to deformities and physical disabilities, often irreversible [1–3].

The World Health Organization (WHO) has defined leprosy as a public health problem, especially in countries whose prevalence rate exceeds one case per 10,000 inhabitants. In 2020, approximately 127,396 new cases of the disease were reported worldwide, of which 19,195 (15.1%) occurred in the Americas continents; Brazil reported 17,979 cases corresponding to 93.6% of the number of new cases in the Americas [1, 2].

Neurological injury determines sensory and motor alterations that lead to the installation of varying degrees of physical disabilities, among them are plantar ulcers that due to the difficulty of healing, these lesions constitute a great challenge among health professionals who deal with this issue [2, 4–6].

Plantar ulcers are the most common severe disabilities resulting from leprosy and can occur in the forefoot, midfoot or hindfoot, in approximately 10 to 30% of all patients affected by the disease [7, 8].

These lesions do not heal easily and are resistant to conventional dressings [5] due to two main causes: the presence of persistent deep infection and the presence of plantar bony prominences due to the disintegration of bone structure caused by neurological injury. Areas of ischemia caused by bony prominences are non-healing factors [9–11].

We found in the medical literature several alternatives for the management of leprosy chronic plantar ulcer varying from conservative treatment unto surgical approach. Conservative treatment ranges from the local use of various antibiotics, antiseptics [12] chemicals, biologicals and plant extracts; [13] use of vasodilators, autolytic and biochemical debridements, [14]; growth factors [15]; full contact plaster [16, 17] custom insoles and therapeutic footwear [18], topical insulin [5] and among other forms of dressings and products.

However, a study by Noordeen and Srinivasan demonstrated that chronic plantar ulcers with associated bony prominences and that were treated conservatively with dressings, rest or full contact plaster, relapsed around one month after healing [19], and recent study by Guimarães et al, (2019) on the various therapies used for dressings on leprosy ulcers, shows that there is no strong evidence that conventional therapies for treating leprosy ulcers decrease the disabling sequelae caused by the disease [20].

Due to the high rate of recurrence of these lesions by conservative methods, several surgeons around the world have reported their techniques with their advances in the use of grafts

and skin flaps with success on closure of the lesions, but also have some limitations in relation to the high rates of recurrences of the lesions. Surgical methods consist of several techniques such as: surgical debridement [13, 21]; use of skin grafts [8]; posterior tibial nerve neurolysis [22, 23]; use of bipediculated local flaps [10]; transposition of plantar flaps with blood supply and neurological plexus [24, 25]; autologous follicular and smashed dermal graft [26–28].

It is important to highlight that the various surgical techniques for closing plantar ulcers do not include the correction of plantar bone fragments and these are predisposing factors for their appearance and chronicity. The main focus of these techniques is only the debridement of infected tissues followed by the application of grafts and skin flaps [8–10, 13, 22–29].

The differential of the technique highlighted in this study is the orthopedic surgeon's view of implementing corrective resection of bone prominences in the plantar region when they are present. These prominences are visualized on foot radiographs in anteroposterior and lateral incidences.

The present study aimed to evaluate the epidemiological profile of patients with chronic plantar ulcers with healing difficulty when conventional dressings are used, as well as to evaluate the results of orthopedic surgical treatment with the bipediculated flap technique, however, giving importance to the resection of bony prominences and the realignment of the plantar anatomical architecture of the foot.

## Materials and methods

### Study design

This is a retrospective cross-sectional study of medical records of patients with chronic plantar ulcers operated at the Adriano Jorge Foundation Hospital from 2008 to 2018, with a non-probabilistic sample, obtained for convenience, corresponding to the target population assisted in an outpatient service at Colonia Antonio Aleixo neighborhood in the city of Manaus.

### Area of study, recruitment and selection of patients

Patients with leprosy chronic plantar ulcer and the presence of plantar bony prominences evident on radiographs of the foot in anteroposterior (AP) and profile (P) incidences, attended at the Antônio Aleixo Polyclinic and submitted to orthopedic surgical treatment at the Adriano Jorge Foundation Hospital in the city of Manaus.

**Inclusion criteria.** Patients older than 18 years of age, of both sexes with leprosy chronic plantar ulcers not responsive to conventional dressings, with the presence of bony prominences in the lesion bed.

**Exclusion criteria.** Patients with bone pathology and non-leprosy peripheral neuropathy associated; presence leprosy reactions and use of polychemotherapy; foot radiographs in anteroposterior and lateral incidences without the presence of plantar bony prominences.

Eligible patients were invited to participate by completing the Informed Consent Form and at that time the following preoperative tests were requested: blood count, coagulogram, urea, creatinine, sodium, potassium, glycemia, electrocardiogram and chest X-ray; These tests were evaluated by the cardiologist for surgical risk.

### Surgical procedures and rehabilitation

### Adriano Jorge Foundation Hospital

**Surgical technique.** Patient in supine position under spinal anesthesia and sedation; Antisepsis and asepsis; placement of sterile fields; identification of plantar ulcer; intralesional incision, debridement of devitalized and or infected tissues; identification of bony prominences in

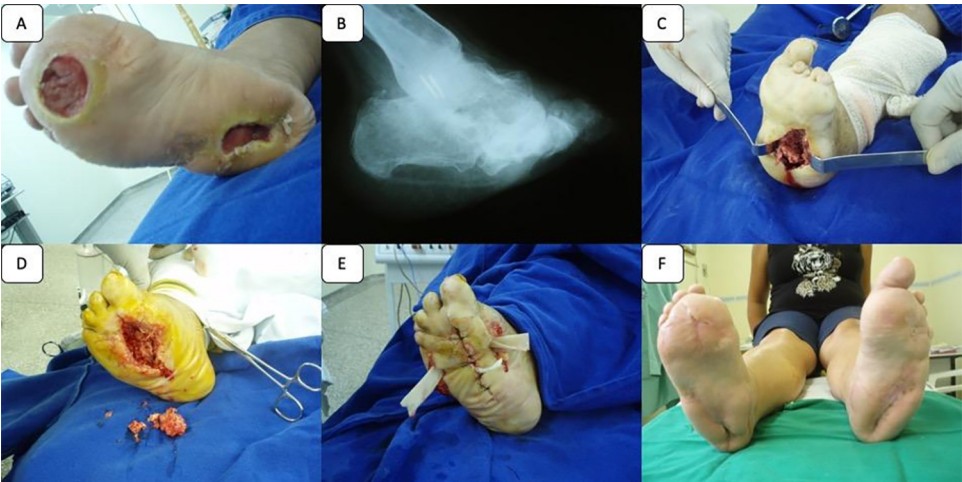

**Fig 1.** Surgical technique of plantar ulcer: **A**—Chronic Plantar Ulcer; **B**—Radiography of the foot with bony prominences; **C**—Resection of bony prominences; **D**—Debridement of devitalized and necrotic soft tissues; **E**—Cutaneous flaps bipediculated by medial, lateral and plantar incision; F- Total closure of chronic plantar ulcer.

the lesion bed with the aid of the imaging intensifier apparatus (fluoroscopy); corrective resection of plantar bony prominences; radiographic control to make sure of satisfactory bone resection; lateral and medial incision of the foot for the advancement of the local bipedicular flap; primary closure of the plantar ulcer; with opening of the lateral and medial incisions; Introduction of lateral and medial drain; occlusive dressing and lateral and medial incisions are left open for healing by second intention (Fig 1).

After post-operative clinical stabilization, around second or third day, the patient is transferred to the Geraldo da Rocha Hospital for postoperative follow-up.

## Dr. Geraldo da Rocha General Hospital

In this hospital unit occurs medical follow-up and postoperative dressings are exchanged by the trained and capable nursing staff. The permanence of the patient in this hospital unit occurs until the complete healing of the lesions, motivated by the financial needs of the patients in terms of locomotion to perform outpatient dressings, as well as by the safety of clinical follow-up.

## Colônia Antônio Aleixo rehabilitation center

It is a public health unit (orthopedic workshop) for the manufacture of orthotics and prostheses where the customization of insoles and therapeutic footwear is performed for accommodation and protection of healed feet.

## Outpatient follow-up

### Colônia Antônio Aleixo polyclinic

After the preparation of the shoes and insoles, the patients return to the Colônia Antônio Aleixo Polyclinical outpatient service for clinical follow-up, by the multidisciplinary team.

## Statistical analysis

The clinical and epidemiological data of the study participants were tabulated and organized into spreadsheets (Excel 2010). Data analysis was performed using Stata v13.0 statistical software. The clinical and epidemiological profile of the studied population is presented with the absolute and relative frequency of the studied population. Pearson's chi-square test ($\chi^2$) was performed to evaluate the correlation between ulcer closure time (in weeks) and the variables studied. A 95% confidence interval (95% CI) and p- value was considered $\leq 0.05$.

**Variables studied.** Gender, age, schooling, marital status, ulcer time, laterality, use of footwear and insoles, quality of dressings, comorbidities and healing time in weeks.

## Ethical considerations

The Research Ethics Committee of the FMT-HVD (Tropical Medicine Foundation—Dr. Heitor Vieira Dourado) approved this project: CAAE 49529415.5.0000.0007, Approval number: 1,322,913.

## Results

### 1-Social-epidemiological aspects

The study included 234 patients (295 feet) who underwent surgery in the from 2008 to 2018. Of these, 129 (55.1%) were male and 105 (44.9%) were female. The average age was 69.5 (D.P. ± 6.7) years; 218 (93.16%) were over 60 years old. Regarding schooling, 194 (82.9%) were functionally illiterate (Tabela1).

### 2-Clinical aspects before the surgical procedure

The majority of patients, 207 (88.4%), had plantar ulcers, for a period over 10 years, and 42 of which (17.9%) had the above-mentioned lesions for more than 30 years; In regarding laterality, 173 (73,9%) were unilateral and 61 (26,1%) were bilateral.

Only 58 (24.8%) used appropriate insoles and footwear, in relation to dressings, 142 (60.7%) were performed by nurse aides in health center facilities; 73 (31.2%) performed dressings in their own homes and 19 (8.1%) of the patients performed dressings both at home and at health center facilities. As to the comorbidities, 62 (26.5%) had systemic arterial hypertension (SAH),12(5,1%) the patients had cardiopathies; 7(3.0%); 7(3%) were obese. 120(51.3%) were smokers (Table 1).

### 3-Results after surgical procedure and statistical correlation

After surgical treatment, it was found that total wound healing occurred in an average time of 12 weeks; In the statistical analysis, the factors that contributed to shorter healing time were: patients of lower age group (P = 0.0011), regular use of footwear and orthopedic insoles before surgery (P = 0.009) and dressings performed during the period of ulcer opening by nurse aides in health center facilities (P = 0.0021).

Obesity was a factor with statistical significance that correlated with the delay of the healing time of the plantar ulcer (P = 0.0024) (Table 1).

## Discussion

The overall incidence of leprosy disease remains high and patients often have severe complications associated with the disease, including chronic plantar ulcers [1, 8].

**Table 1. Epidemiological profile and statistical correlation of socioepidemiological variables with healing time, in weeks, of patients undergoing surgical intervention by chronic plantar ulcers secondary to leprosy, from 208 to 2018 in the city of Manaus, Amazon–Brazil.**

| | TOTAL | UNTIL 8 WEEKS | 8 WKS. TO 12 WKS. | > 12 SEM. | P- VALUE |
|---|---|---|---|---|---|
| | N = 234 | N = 25 | N = 140 | N = 69 | |
| **Gender** | | | | | 0.12 |
| Male | 129/234 (55.1%) | 11/25 (44.0%) | 81/140 (57.9%) | 37/69 (53.6%) | |
| Female | 105/234 (44.9%) | 14/25 (56.0%) | 59/140 (42.1%) | 32/69 (46.4%) | |
| **Age (group)** | | | | | 0.011 |
| **51 a 60** | 16/234 (6.8%) | 1/25 (4.0%) | 12/140 (8.6%) | 3.69 (4.3%) | |
| **61 a 70** | 115/234 (49,1%) | 10/25 (40.0%) | 65/140 (46.4%) | 40/69 (58.0%) | |
| **71 a 80** | 85/234 (36,3%) | 13/25 (52,0%) | 51/140 (36.4%) | 21/69 (30.4%) | |
| **> 80** | 18/234 (7.7%) | 1/25 (4.0%) | 12/140 (8.6%) | 5/69 (7.2%) | |
| **Education** | | | | | 0.063 |
| Illiterate | 55/234 (23.5%) | 3/25 (23.5%) | 32/140 (22.9%) | 20/69 (29.0%) | |
| Incomplete Elementary School | 139/234 (59.4%) | 15/25 (60.0%) | 87/140 (62.1%) | 37/69 (53.6%) | |
| Elementary School | 23/234 (9.8%) | 5/25 (20.0%) | 11/140 (7.9%) | 7/69 (10.1%) | |
| High School or More | 17/234 (7.3%) | 2/25 (8.0%) | 10/140 (7.1%) | 5/69 (7.2%) | |
| **Marital Status** | | | | | 0.30 |
| Single | 116/231 (50.2%) | 11/25 (44.0%) | 75/138 (54.3%) | 30/68 (44.1%) | |
| Married | 72/231 (31.2%) | 8/25 (32.0%) | 43/138 (31.2%) | 21/68 (30.9%) | |
| Divorced | 3/231 (1.3%) | 1/25 (4.0%) | 2/138 (1.4%) | 0/68 (0.0%) | |
| Widowed | 40/231 (17.3%) | 5/25 (20.0%) | 18/138 (13.0%) | 17/68 (25.0%) | |
| **Time with Ulcer** | | | | | 0.063 |
| < 10 | 27/234 (11.5%) | 2/25 (8.0%) | 18/140 (12.9%) | 7/69 (10.1%) | |
| 11 a 20 | 99/234 (42.3%) | 9/25 (36.0%) | 55/140 (39.3%) | 35/69 (50.7%) | |
| 21 a 30 | 66/234 28.2(%) | 8/25 (32.0%) | 45/140 (32.1%) | 13/69 (18.8%) | |
| > 30 | 42/234 (17.9%) | 6/25 (24.0%) | 22/140 (15.7%) | 14/69 (20.3%) | |
| **Compromised foot** | | | | | 0.87 |
| Left | 91/234 (38.9%) | 11/25 (44.0%) | 52/140 (37.1%) | 28/69 (40.6%) | |
| Right | 82/234 (35.0%) | 7/25 (28.0%) | 50/140 (35.7%) | 25/69 (36.2%) | |
| Bilateral | 61/234 (26.1%) | 7/25 (28.0%) | 38/140 (27.1%) | 16/69 (23.2%) | |
| **Use of Shoes or Insole** | 58/234 (24.8%) | 6/25 (24.0%) | 34/140 (24.3%) | 18/69 (26.1%) | **0.009** |
| **Dressing** | | | | | 0.021 |
| At home | 73/234 (31.2%) | 8/25 (32.0%) | 45/140 (32.1%) | 20/69 (29.0%) | |
| Outpatient Health | 19/234 (8.1%) | 3/25 (12.0%) | 7/140 (5.0%) | 9/69 (13.0%) | |
| Home care and Outpatient | 142/234 (60.7%) | 14/25 (56.0%) | 88/140 (62.9%) | 40/69 (58.0%) | |
| **Comorbidity** | | | | | |
| Cardiopathy | 12/234 (5.1%) | 1/25 (4.0%) | 5/140 (3.6%) | 6/69 (8.7%) | **0.59** |
| Obesity | 7/234 (3.0%) | 0/25 (0.0%) | 1/140 (0.7%) | 6/69 (8.7%) | **0.024** |
| Hypertension | 62/234 (26.5%) | 4/25 (16.0%) | 38/140 (27,1%) | 20/69 (29.0%) | **0.032** |
| Smoking | 120/234 (51,3%) | 12/25 (48.0%) | 72/140 (51,4%) | 36/69 (52.2%) | **0.82** |

In our sample, the sociodemographic profile shows a predominance of males (55.1%) with an average age of 69.5 years, findings that corroborate the scientific literature, justified by the higher exposure of male patients in relation to females and the delay in the diagnosis of the disease [25, 30–33].

The majority of the participants are functional illiterate, demonstrating the social characteristics of isolation and neglected disease, present mainly in the low-income population [1, 2, 30, 33].

One point that caught our attention in our study is the length of time that these plantar lesions were open, which the majority was longer than 10 years despite conventional dressings

were done. Similar findings were observed in several published studies demonstrating the great challenge in the management of these lesions [9, 32, 33].

Despite the great advance in the last decades of several surgical techniques used for the closure of chronic plantar ulcers in leprosy disease, there is a high recurrence rate ranging from 18.35 to 90% of cases [8, 13, 25] explained in part by the understanding of the pathophysiology of plantar ulcers where bone prominences, even if not infected, are a predisposing factor in the appearance of same [9, 10].

These orthopedic deformities are not addressed in the surgical time in the expressive majority of surgical techniques, focusing only on the debridement of infected tissues and then in the cutaneous closure of these lesions [8, 13, 21–28].

It is scientific knowledge that leprosy neuropathy triggers a process of disintegration and multifragmentation of the foot bone structure. This anatomical destructuring results in the appearance of bony prominences in the plantar area of the feet, being an important factor in the appearance and chronicities of these lesions [9, 10, 24, 32].

Jin et al., 2009 affirm that the surgical treatment of chronic plantar ulcers in leprosy disease should include the relief of the plantar high-pressure zone through bone resection and flap transposition, providing good therapeutic effect in the short and long term [21].

However, we found few studies in the world literature that show short, medium and long-term results using the technique of corrective resection of uninfected plantar bony prominences, and that for this reason this work aims to fill this gap in this branch of knowledge.

These bone alterations are visualized on radiographs (X-ray) of the foot at anteroposterior and lateral incidence in the pre-surgery period. Applying the proposed technique, total closure of plantar lesions was observed in 100% of the cases in the average time of 12 weeks. With these results, we can see the importance of orthopedic evaluation in the management of these difficult-to-heal lesions.

In the statistical analysis, it was observed that the dressings that were performed in the pre-surgical procedure period by nurse aides in health center facilities presented shorter healing time comparing those dressings performed by patients at home (P = 0.021), corroborating the work of Mustapha et al. in which it is demonstrated that the appearance of plantar ulcers can be avoided through dressings performed by trained health professionals, because they contribute to the preservation of viable tissues in the lesion bed [34].

Another determining factor for shorter healing time was the regular use of orthopedic footwear and insoles (P = 0.024), in fact, studies published in the world literature report that the healing of plantar ulcers requires measures of self-care and the correct use of orthotics and protective devices that include footwear and insoles [35, 36].

These devices reduce pressure peaks in the plantar area avoiding the appearance of ulcers and providing them for free reduces the costs of leprosy disease treatment in developing countries [37, 38].

The lower age group (P = 0.001) also obtained a positive correlation in the shorter healing time explained by the patient's clinical conditions in relation to self-care [35].

Obesity contributed to the delay in the healing time of post-procedure injuries (P = 0.024) corroborating studies by Upputuri et al. (2020), considering the systemic inflammatory state favored by this clinical condition [32].

## Future prospects

As future perspectives there is a need for a second evaluation in the search for long-term results to determine the recurrence rate of the lesions and evaluate the impact on quality of life and postoperative personal satisfaction.

## Study limitations

As a study limitation, we did not have the opportunity to gather anthropometric measurements of these patients, such as height and weight, as well as the diameter and depth of plantar ulcers.

## Conclusion

In the present study, it was observed that patients with chronic leprosy plantar ulcers are mostly male and with low school level. The regular use of shoes and insoles and dressings performed by nurse aids in health center facilities in the pre-surgery period contributed to shorter healing time of the lesions after surgery.

Orthopedic surgical treatment with corrective resection of bony prominences associated with the bipediculated local cutaneous flap proved to be an efficient therapeutic method for the closure of chronic plantar ulcers. This is a reproducible method with excellent results justifying the importance and contribution of the orthopedic surgeon in the context of the multidisciplinary team for evaluation and handling these complex lesions.

## Supporting information

**S1 Data. Demographic and health data of patients submitted to surgical intervention due to chronic plantar ulcer secondary to leprosy, between 2008 and 2018 in the city of Manaus, AM.**
(XLSX)

## Acknowledgments

The authors thank the administrators and employees of the Doutor Heitor Vieira Dourado Tropical Medicine Foundation, Adriano Jorge Hospital Foundation, Antônio Aleixo Policlinic and Geraldo da Rocha Hospital, for their support in obtaining medical documents and data. We also thank the Tropical Medicine Graduate Program (PPGMT/UEA) and the Amazonas State Research Support Foundation (FAPEAM) for supporting and funding the expenses regarding publication of this paper under Resolution N. 002/2008, 007/2018 and Notice N.005/2019–PRÓ-ESTADO. Likewise, we thank all patients who participated in the treatment procedure, that without them this study would not have been possible.

## Author Contributions

**Conceptualization:** Francisco Mateus João, Silmara Navarro Peninni, Mônica Regina Hosannah da Silva e Silva, Jorge Augusto de Oliveira Guerra.

**Data curation:** Francisco Mateus João, Arineia Soares da Silva, Katia do Nascimento Couceiro, Mônica Regina Hosannah da Silva e Silva, Marcello Facundo do Vale Filho, Guilherme Miranda Silva de Oliveira, Lucas Silva Ferreira, Victor Irungu Mwangi, Maria das Graças Vale Barbosa Guerra.

**Formal analysis:** Francisco Mateus João, Katia do Nascimento Couceiro, Alba Regina Jorge Brandão, Mônica Regina Hosannah da Silva e Silva, Victor Irungu Mwangi, Maria das Graças Vale Barbosa Guerra, Jorge Augusto de Oliveira Guerra.

**Investigation:** Francisco Mateus João, Silmara Navarro Peninni, Alba Regina Jorge Brandão, Victor Irungu Mwangi, Bernardo Maia da Silva, Maria das Graças Vale Barbosa Guerra.

**Methodology:** Francisco Mateus João, Maria das Graças Vale Barbosa Guerra, Jorge Augusto de Oliveira Guerra.

**Project administration:** Francisco Mateus João, Alba Regina Jorge Brandão, Maria das Graças Vale Barbosa Guerra.

**Resources:** Francisco Mateus João, Mônica Regina Hosannah da Silva e Silva, Maria das Graças Vale Barbosa Guerra, Jorge Augusto de Oliveira Guerra.

**Software:** Arineia Soares da Silva, Maria das Graças Vale Barbosa Guerra, Jorge Augusto de Oliveira Guerra.

**Supervision:** Francisco Mateus João, Arineia Soares da Silva, Maria das Graças Vale Barbosa Guerra, Jorge Augusto de Oliveira Guerra.

**Validation:** Francisco Mateus João, Alba Regina Jorge Brandão, Maria das Graças Vale Barbosa Guerra, Jorge Augusto de Oliveira Guerra.

**Visualization:** Francisco Mateus João, Maria das Graças Vale Barbosa Guerra.

**Writing – original draft:** Francisco Mateus João, Silmara Navarro Peninni, Zanair Soares Vasconcelos, Arineia Soares da Silva, Katia do Nascimento Couceiro, Alba Regina Jorge Brandão, Marcello Facundo do Vale Filho, Guilherme Miranda Silva de Oliveira, Maria das Graças Vale Barbosa Guerra, Jorge Augusto de Oliveira Guerra.

**Writing – review & editing:** Francisco Mateus João, Silmara Navarro Peninni, Zanair Soares Vasconcelos, Arineia Soares da Silva, Mônica Regina Hosannah da Silva e Silva, Victor Irungu Mwangi, Bernardo Maia da Silva, Maria das Graças Vale Barbosa Guerra, Jorge Augusto de Oliveira Guerra.

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
