## [Decision Letter · Decision Letter 0]

21 Sep 2022

PONE-D-22-04569EPIDEMIOLOGICAL CLINICAL PROFILE AND CLOSURE OF CHRONIC PLANTAR ULCERS IN PATIENTS WITH LEPROSY SEQUELAE UNDERGOING ORTHOPEDIC SURGERY IN A CITY IN WESTERN AMAZONIAPLOS ONE

Dear Dr. João,

Thank you for submitting your manuscript to PLOS ONE. After careful consideration, we feel that it has merit but does not fully meet PLOS ONE’s publication criteria as it currently stands. Therefore, we invite you to submit a revised version of the manuscript that addresses the points raised during the review process.

We look forward to receiving your revised manuscript.

Kind regards,

Yaodong Gu

Academic Editor

PLOS ONE

Journal Requirements:

“NO”

d)        If you did not receive any funding for this study, please state: “The authors received no specific funding for this work

Additional Editor Comments (if provided):

The methods part shall be more detailing to express the how you complete the experiment.

Reviewers' comments:

Reviewer's Responses to Questions

**Comments to the Author**

1. Is the manuscript technically sound, and do the data support the conclusions?

Reviewer #1: Partly

Reviewer #2: Yes

Reviewer #3: Yes

2. Has the statistical analysis been performed appropriately and rigorously? 

Reviewer #1: Yes

Reviewer #2: Yes

Reviewer #3: Yes

3. Have the authors made all data underlying the findings in their manuscript fully available?

Reviewer #1: Yes

Reviewer #2: Yes

Reviewer #3: No

4. Is the manuscript presented in an intelligible fashion and written in standard English?

Reviewer #1: No

Reviewer #2: Yes

Reviewer #3: Yes

5. Review Comments to the Author

Reviewer #1: The authors sought to describe the clinical, epidemiological profile and the results of orthopedic surgical treatment of chronic and relapsing foot ulcers in leprosy patients, the purposes of this study seem to be clear, however, several serious issues must be addressed, and it is suggested that the authors should consider all these comments before submitting it for peer review.

1. For the abstract session, 1) the authors should briefly explain the significance of this study before giving the purposes; 2) more details about the methods and main findings of this study should be presented here.

2. For the introduction session, 1) I could not see the gap that the authors were going to bridge, nor can I see the novelty of this study. Did the authors aim to verify the effects of the surgical orthopedic treatment proposed by Hidalgo and Shaw in the last century (1986)? Numerous studies have investigated the novel surgical treatments for plantar ulcers in leprosy, thus what are the main novelty and significance of this study? It is suggested that the authors must consider making a throughout review of previous studies and consider rewriting this part to further highlight the novelty and significance; 2) the authors should consider further strengthening the logical structure of this part, making clear explanations of the definition, symptoms, treatments of foot ulcers in leprosy patients; 3) update the statistical data and their corresponding refs in this part.

3. for the method section, the author must explain the method in detail. For example, 1) no details about the subject recruitment criteria; 2) no details about the clinical profile used for evaluation in this study; 3) it is hard to understand what the authors tried to explain in the two paragraphs below Figure 1. These contents were also certainly not details about the patient recruitment and selection; 4) it is suggested that the authors should further explain the statistical methods applied for this study.

4. For the results session, 1) it is suggested that the authors should include some sub-sessions here for better description; 2) the results are duplicates in the results session, on the tables, and in the discussion session. I recommend the authors keep just the table on the results table and make the discussion stronger. The text on the results session with the description of the table is not helping the reader.

5. For the discussion session, few deep comparisons were made, and no practical information was presented based on the findings of this study. It is suggested that the authors should make more comparisons with previous related studies and explain the limitations of this study.

6. For the conclusion session, it is suggested that this part should be further strengthened based on the main findings of this study.

Reviewer #2: Review comment

This manuscript entitled “Epidemiological Clinical Profile and Closure of Chronic Plantar Ulcers in Patients with Leprosy Sequelae Undergoing Orthopedic Surgery in a City in Western Amazonia” primarily aimed to investigate the epidemiological profile of patients with chronic plantar ulcers with difficulty in healing, secondary to leprosy. The results of this study provide guidance for public health and clinical medicine. While it is a very interesting topic. But I think this manuscript has some flaws to fill in before it can be published in a journal. There are several questions should be addressed, which list below. I give a minor revision for this manuscript.

Specific comments

1. In the Abstract part, in the opinion of reviewer, the author provided too much descriptions of background, which may be too long-winded. I suggest that the authors provide more descriptions of conclusion of this study in the abstract part.

2. “A total of 234 patients (295 feet) underwent surgical treatment of ulcers: 129 (55.1%) were male; mean age was 69.5 years.” Please provide more anthropometric information about these patients, such as Height, weight, length of illness.

3. In the introduction part.“It may present slow and progressive evolution, leading to deformities and physical disabilities, often irreversible.” Please add some references to support this sentence.

4. “Nerve damage caused by Mycobacterium leprae induces sensory and motor alterations that lead to the installation of varying degrees of a physical disability that can interfere with patients’ social and economic life, resulting in stigma and discrimination.” This sentence is too general and unclear. Can you be more specific about what you mean for nerve damage.

5. “This change in the foot bone arrangement then causes a maladjusted gait in the patient, leading to new pressure points in non-appropriate parts of the foot.” Which parts of the foot.

6. “These lesions must be addressed and adequately prevented, as they constitute entry points for infections that can progress to severe complications, such as osteomyelitis, plantar ulcers, and even limb amputations.” Please use more than one references to prove this statement.

7. hat is the author's research hypothesis, which I suggest to be added to the last paragraph of the introduction?

8. In the Methodology part.” After healing, new radiographs were taken to control and request the manufacture of customized insoles and footwear, aiming at accommodating and protecting the feet.” What is the basis for the customization of insoles and footwear?

9. In the Results part. In the opinion of the reviewer, the pixel in Figure 1 is too blurred, please replace a clearer figure.

10. In the Discussion part. What are the limitations of this study? Please provide relevant description.

11. In the Conclusion part. In the opinion of the reviewer, the description in the conclusion part was too verbose, and the reviewer suggests that the authors should abbreviate the section and focus on the main findings of this study.

Reviewer #3: This is a well written artilce that addresses an important issue. However, I do have some comments about the article in its present form

1. There are a number questions at the start of the article which have not been properly answered. As an example for financial disclosure the response was "NO". The notes clearly state the appropriate words to be used in this section.

2, As above for Data availability. There is no indication of where these data may be accessed.

3. Abstract. A total of 234 patients... in contrast 208 patients.....I do not understand what this paragraph means.

4. I am confused about the results at the end of page 7 and start of page 8. There are two statistically significant p-values but I am uncertain what was being compared. The statement "The age of the participants were significant variables.." is meaningless without a statement of which groups were being compared.

5, Page 8 (last sentence)... "all 295 feet had complete healing..." .Can the authors confirm that there were no drop outs and all were followed until complete healing. This is an elderly group and I am astounded that all survived and healed with no drop outs.

6. Table 3. You seem to have 100% recording of all variables used in this study. Is this correct?

7 Table 3. Please give the statistical method in the table heading. For a correlation it would be helpful to give the correlation coefficients for each p-value.

8. i presume that Genre means Gender in table 1. Please change

6. PLOS authors have the option to publish the peer review history of their article (what does this mean?). If published, this will include your full peer review and any attached files.

Reviewer #1: **Yes: **Song Yang

Reviewer #2: No

Reviewer #3: **Yes: **Peter J Franks

---

## [Author Response · Author response to Decision Letter 0]

30 Nov 2022

RESPONSE TO REVIEWERS

PONE-D-22-04569

EPIDEMIOLOGICAL CLINICAL PROFILE AND CLOSURE OF CHRONIC PLANTAR ULCERS IN PATIENTS WITH LEPROSY SEQUELAE UNDERGOING ORTHOPEDIC SURGERY IN A CITY IN WESTERN AMAZONIA

Journal Requirements:

Response: We fully reviewed the manuscript and followed the instructions according to Guidelines.

Response: We emphasized in the Materials and Methods section that all participants signed an Informed Consent Form (page 6, lines 133 and 134).

d) If you did not receive any funding for this study, please state: “The authors received no specific funding for this work.

Response: The author(s) received no specific funding for this work. 

Additional Editor Comments (if provided):

The methods part shall be more detailing to express the how you complete the experiment.

Response: The Materials and Methods section was fully updated in order to improve reading and chronologically organize the steps followed, including the recruitment criteria (pages 5-8, lines 117 to 144 and 153 to 169).

Reviewer #1: The authors sought to describe the clinical, epidemiological profile and the results of orthopedic surgical treatment of chronic and relapsing foot ulcers in leprosy patients, the purposes of this study seem to be clear, however, several serious issues must be addressed, and it is suggested that the authors should consider all these comments before submitting it for peer review.

1. For the abstract session, the authors should briefly explain the significance of this study before giving the purposes; more details about the methods and main findings of this study should be presented here.

Response: The abstract was rewritten in order to attend all suggestion given by the three reviewers (page 2 – lines 24 to 43).

2. For the introduction session, I could not see the gap that the authors were going to bridge, nor can I see the novelty of this study. Did the authors aim to verify the effects of the surgical orthopedic treatment proposed by Hidalgo and Shaw in the last century (1986)? 

Numerous studies have investigated the novel surgical treatments for plantar ulcers in leprosy, thus what are the main novelty and significance of this study? It is suggested that the authors must consider making a throughout review of previous studies and consider rewriting this part to further highlight the novelty and significance;

Response: Regarding the description of the technique used, there was a mistake: in fact, the technique used was that of Srinivasan and not of Hidalgo and Shaw, as previously described.

Regarding the literature review, the following text was added (included in the Introduction section on pages 4- 5, lines 87 to 110):

“We found in the medical literature several alternatives for the management of chronic plantar ulcer from conservative to surgical treatment. Conservative treatment ranges from the local use of a variety of antibiotics, antiseptic agents, chemical, biological and vegetal extracts; use of vasodilators, autolytic and biochemical debridements, growth factors, total contact plaster and customized therapeutic insoles and footwear among other dressing forms and products [16-23].

However, studies by Noordeen and Srinivasan demonstrated that chronic plantar ulcers with bony prominences associated with an inflammatory process between the friable tissue and the bone and which were treated conservatively by dressings, rest or total contact plaster, recur around one month after healing [24].

Due to the high rate of recurrence of these lesions by conservative methods, several surgeons around the world have reported their techniques with their advances and limitations. Methods such as: a) surgical debridements [22, 25, 26]; b) use of skin grafts from the dorsum of the foot [27]; c) posterior tibial nerve neurolysis [28, 29]; d) use of local bipedicled flaps [30]; e) plantar flaps with blood and neurological plexus supply [25, 30, 31].

It is important to emphasize that the several surgical techniques to close plantar ulcers do not contemplate the correction of plantar bone fragments that are predisposing factors for the appearance of plantar ulcers as well as their recurrence [7, 11, 30, 31]. 

We found that these techniques consist primarily in the debridement of infected tissues and the closure of the ulcer pure and simple. The bipedicle flap technique proposed by Srinivasan mobilizes the flap around the ulcer and was the technique used by the author differently from the techniques presented, the author associates the technique with plantar bone resection in cases where there is presence of these bone lesions (identified in the foot radiographs in the anteroposterior and profile views).”

 2.2) the authors should consider further strengthening the logical structure of this part, making clear explanations of the definition, symptoms, treatments of foot ulcers in leprosy patients; 

Response: This review was done in detail and are described in the restructured paragraphs of the introduction (page 4, lines 80 to 86):

“Price in 1964 introduced the term "perforating ulcer" as a chronic ulceration of an anesthetic region of the foot related to areas of bony prominences, resistant to local or systemic treatment and characterized by a high tendency to recur [11].

Srinivasan defines chronic ulcer as the ulcer that does not heal easily due to two main causes: a) presence of deep and persistent infection or b) the presence of plantar bony prominences due to disintegration of bony framework caused by neurological injury. Areas of ischemia caused by bony prominences are factors of non-healing [10,15].”

3) update the statistical data and their corresponding refs in this part.

Response: The epidemiological profile was updated with data from 2021 and 2022. These informations were added to the manuscript on page 3, lines 57 to 68.

“The World Health Organization (WHO) has defined leprosy as a public health problem, especially in countries whose prevalence rate exceeds one case per 10,000 inhabitants. In the year 2020, about 127,396 new cases of the disease were reported in the world, of these, 19,195 (15.1%) occurred in the Americas region, with Brazil with 17,979 cases corresponding to 93.6% of the number of new cases in the Americas [3,5].

The neurological lesion determines sensory and motor alterations that lead to the installation of varying degrees of physical disabilities (WHO-DG), among them are the plantar ulcers that due to the difficulty in healing these lesions constitute a major challenge among health professionals who deal with this issue [6-8].

Plantar ulcers are the most common severe disabilities due to leprosy and occur in approximately 10 to 20% of all patients affected by the disease, despite recent advances in multidrug therapy [9-11].”

3. for the method section, the author must explain the method in detail. For example, 

3.1) no details about the subject recruitment criteria; 

Response: The Materials and Methods section was fully updated in order to improve reading and chronologically organize the steps followed, including the recruitment criteria (pages 5-8, lines 117 to 144 and 153 to 169).

3.2) no details about the clinical profile used for evaluation in this study; 

Response: Clinical profile for evaluation of recruitment were included. 

Pages 5-6, lines 123 to 137

“Patient population and study area 

Patients with chronic plantar ulcer with presence of bony prominences in the radiographic study of the foot of leprosy origin seen at the Policlínica Antônio Aleixo and submitted to orthopedic surgical treatment at Adriano Jorge Foundation Hospital.

Patient recruitment and selection 

Polyclinic Colônia Antônio Aleixo 

Patients over 18 years of age, of both genders, who presented chronic plantar ulcers unresponsive to conventional dressings, with presence of bony prominences in the bed of the lesion identified by X-ray in the anteroposterior (AP) and profile (P) incidence were eligible for inclusion in this work. 

These patients were invited to participate by filling out the Informed Consent Form (ICF) and at that moment the following preoperative exams were requested: blood count, coagulogram, urea, creatinine, sodium, potassium, glycemia, electrocardiogram, and AP/P chest X-ray. These exams are taken to the cardiologist in the same unit for the preoperative risk.”

3.3) it is hard to understand what the authors tried to explain in the two paragraphs below Figure 1. These contents were also certainly not details about the patient recruitment and selection; 

Response: Below Figure 1, we included the specifications of the local health services where the patients were transferred after the surgical procedure (page 7, lines 153 to 163). 

3.4) it is suggested that the authors should further explain the statistical methods applied for this study.

Response: Statistical analysis was revised and rewritten. The correction is found on page 8, lines 170 to 181).

4. For the results session, 

4.1) it is suggested that the authors should include some sub-sessions here for better description; 

Response: We organized chronologically the results obtained as well as kept only one table with all the variables and the p-values (pages 8-10, lines 188 to 222).

4.2) the results are duplicates in the results session, on the tables, and in the discussion session. I recommend the authors keep just the table on the results table and make the discussion stronger. The text on the results session with the description of the table is not helping the reader.

Response: We excluded Tables 1 and 2, that indeed were duplicates and kept all the relevant informations in Table 1 (page 10). 

5. For the discussion session, few deep comparisons were made, and no practical information was presented based on the findings of this study. It is suggested that the authors should make more comparisons with previous related studies and explain the limitations of this study.

Response: In order to readjust and provide new references for the Discussion section, we made a major revision on this section, as shown on pages 11-14, lines 224 to 323.

6. For the conclusion session, it is suggested that this part should be further strengthened based on the main findings of this study.

Response: The Conclusion was rewritten as follows:

“In the present study, it was observed that patients with chronic plantar leprosy ulcers are mostly male, with a mean age of 69.5 years. 

The population has low education levels and limited family and social ties, reinforcing the stigma and isolation of these individuals. The period of permanence of the opening of the chronic plantar ulcer was mostly above 10 years, ranging from 0 to 50 years. It was also noted that most of the patients evaluated did not make regular use of insoles and adapted footwear. 

Finally, it was observed that surgical treatment using the local bipedicle skin flap technique associated with resection of bony prominences and corrective osteotomies is an effective therapeutic method for the closure of chronic plantar ulcers. It is a reproducible method with excellent results.”

Reviewer #2: Review comment

This manuscript entitled “Epidemiological Clinical Profile and Closure of Chronic Plantar Ulcers in Patients with Leprosy Sequelae Undergoing Orthopedic Surgery in a City in Western Amazonia” primarily aimed to investigate the epidemiological profile of patients with chronic plantar ulcers with difficulty in healing, secondary to leprosy. The results of this study provide guidance for public health and clinical medicine. While it is a very interesting topic. But I think this manuscript has some flaws to fill in before it can be published in a journal. There are several questions should be addressed, which list below. I give a minor revision for this manuscript.

Specific comments

1. In the Abstract part, in the opinion of reviewer, the author provided too much descriptions of background, which may be too long-winded. I suggest that the authors provide more descriptions of conclusion of this study in the abstract part.

Response: The Abstract section has been rewritten to address this request of improvement. It is shown on page 2, lines 24 to 46.

2. “A total of 234 patients (295 feet) underwent surgical treatment of ulcers: 129 (55.1%) were male; mean age was 69.5 years.” Please provide more anthropometric information about these patients, such as Height, weight, length of illness.

Response Although we find it relevant and important to include the anthropometric measurements so pertinently suggested, it was not the scope of our study. We includeded this as one of the limitations of our work (page 14-15, lines 325 to 329).

3. In the introduction part.“It may present slow and progressive evolution, leading to deformities and physical disabilities, often irreversible.” Please add some references to support this sentence.

Response: Addressing this suggestion, we included the following references (page 3, lines 54 to 56)

1. Souza CDF, Fernandes TRMO, Matos TS, Ribeiro-Filho JM, Almeida GKA, Lima JCB et al. Degree of physical disability in the elderly population affected by leprosy in the state of Bahia, Brazil. Acta Fisiatr. 2017; 24(1): 27-32.

2. Barreto JG, Bisanzio D, Frade MAC, Moraes TMP, Gobbo AR, Guimarães LS et al. Spatial epidemiology and serologic cohorts increase the early detection of leprosy. BMC Inf Dis. 2015; 15(1): 1-9.

4. “Nerve damage caused by Mycobacterium leprae induces sensory and motor alterations that lead to the installation of varying degrees of a physical disability that can interfere with patients’ social and economic life, resulting in stigma and discrimination.” This sentence is too general and unclear. Can you be more specific about what you mean for nerve damage.

Response: Page 3, lines 62 to 65.

“The neurological lesion determines sensory and motor alterations that lead to the installation of varying degrees of physical disabilities (WHO-DG), among them are the plantar ulcers that due to the difficulty in healing these lesions constitute a major challenge among health professionals who deal with this issue [6-8].”

5. “This change in the foot bone arrangement then causes a maladjusted gait in the patient, leading to new pressure points in non-appropriate parts of the foot.” Which parts of the foot.

Response: We included this specific information on page 3, line 66 to 68. The parts of foot are as follows: forefoot, midfoot, hindfoot

6. “These lesions must be addressed and adequately prevented, as they constitute entry points for infections that can progress to severe complications, such as osteomyelitis, plantar ulcers, and even limb amputations.” Please use more than one references to prove this statement.

Response: The following references were added to support the information: 

1. Batista KT, Monteiro GB, Y-Schwartzman UP, Aureliano A, Rosa AG, Correia CZ, et al. Treatment of plantar ulcer due to leprosy. Rev. Bras. Cir. Plast. 2019; 34: 497-503.

2. Tashiro S, Gotou N, Oku Y, Sugano T, Nakamura T, Suzuki H, et al. Relationship between plantar pressure and sensory disturbance in patients with Hansen's disease-Preliminary research and review of the literature. Sensors (Basel). 2020; 20(23): 6976. 

7. What is the author's research hypothesis, which I suggest to be added to the last paragraph of the introduction?

Response: Orthopedic surgical treatment of trophic plantar ulcers with resection of bony prominences and corrective osteotomies, associated with a skin flap, is effective for the complete closure of these lesions.

8. In the Methodology part.” After healing, new radiographs were taken to control and request the manufacture of customized insoles and footwear, aiming at accommodating and protecting the feet.” What is the basis for the customization of insoles and footwear?

Response: Page 8, lines 168.

 Customization is based on complete closure of the plantar ulcer and the patient is advised by the physician to visit the orthopedic workshop for analysis of pressure points through the podoscope to determine the customization of the insole and specific footwear for each patient.

9. In the Results part. In the opinion of the reviewer, the pixel in Figure 1 is too blurred, please replace a clearer figure.

Response: The pictures of Figure 1 were replaced (page 7).

10. In the Discussion part. What are the limitations of this study? Please provide relevant description.

Response: Study Limitations section is shown on page 14, lines 325 to 330.

“We emphasize that this study has limitations where there is a need for a second evaluation in the search for long-term results obtained from the surgical intervention performed and its relationship with the indices of quality and lifestyle of patients, personal satisfaction as well as the recurrence rate of lesions. For this, more anthropometric information about these patients, such as height, weight, length of illness.”

11. In the Conclusion part. In the opinion of the reviewer, the description in the conclusion part was too verbose, and the reviewer suggests that the authors should abbreviate the section and focus on the main findings of this study.

Response: Conclusion section was rewritten and is shown on page 15, lines 333 to 343.

“In the present study, it was observed that patients with chronic plantar leprosy ulcers are mostly male, with a mean age of 69.5 years. 

The population has low education levels and limited family and social ties, reinforcing the stigma and isolation of these individuals. The period of permanence of the opening of the chronic plantar ulcer was mostly above 10 years, ranging from 0 to 50 years. It was also noted that most of the patients evaluated did not make regular use of insoles and adapted footwear. 

Finally, it was observed that surgical treatment using the local bipedicle skin flap technique associated with resection of bony prominences and corrective osteotomies is an effective therapeutic method for the closure of chronic plantar ulcers. It is a reproducible method with excellent results.”

Reviewer #3: This is a well written article that addresses an important issue. However, I do have some comments about the article in its present form.

1. There are a number questions at the start of the article which have not been properly answered. As an example for financial disclosure the response was "NO". The notes clearly state the appropriate words to be used in this section.

Response: The author(s) received no specific funding for this work.

2, As above for Data availability. There is no indication of where these data may be accessed.

Response: These data can be accessed from the author's databases and clinical records at Hospital Geraldo da Rocha, Manaus, Amazonas, Brazil.

3. Abstract. A total of 234 patients... in contrast 208 patients.....I do not understand what this paragraph means.

Response: There was a mistake in writing the data. The review was carried out and the appropriate sentence is: “After the surgical treatment, it was verified that the full healing of the wounds occurred in a maximum period of 20 weeks, with an average time of 12 weeks.”

4. I am confused about the results at the end of page 7 and start of page 8. There are two statistically significant p-values but I am uncertain what was being compared. The statement "The age of the participants were significant variables." is meaningless without a statement of which groups were being compared.

Response: This confusion has been resolved. Table 3 was renamed to Table 1, which contains the p values of each variable. (quote pages and lines) – Method and Results

5, Page 8 (last sentence) ... "all 295 feet had complete healing..." .Can the authors confirm that there were no drop outs and all were followed until complete healing. This is an elderly group and I am astounded that all survived and healed with no drop outs.

Response: Complete healing was assured due to the fact that all patients were hospitalized at Hospital Geraldo da Rocha until the lesions were completely closed, as described in the Methodology. There were no deaths during this period. The maximum period of hospitalization was 20 weeks (5 months).

6. Table 3. You seem to have 100% recording of all variables used in this study. Is this correct?

Response: Table 3 has been renamed to Table 1 – and all variables have full data (100%).

7 Table 3. Please give the statistical method in the table heading. For a correlation it would be helpful to give the correlation coefficients for each p-value.

Response: Frequency comparison analyzes were performed using Pearson's Chi-square test, with no need for coefficient values.

8. I presume that Genre means Gender in table 1. Please change

Response: Gender was modified for Sex. Done.

---

## [Decision Letter · Decision Letter 1]

3 Jan 2023

PONE-D-22-04569R1Epidemiological clinical profile and closure of chronic plantar ulcers in patients with leprosy sequelae undergoing orthopedic surgery in a city in Western AmazoniaPLOS ONE

Dear Dr. João,

Thank you for submitting your manuscript to PLOS ONE. After careful consideration, we feel that it has merit but does not fully meet PLOS ONE’s publication criteria as it currently stands. Therefore, we invite you to submit a revised version of the manuscript that addresses the points raised during the review process.

We look forward to receiving your revised manuscript.

Kind regards,

Yaodong Gu

Academic Editor

PLOS ONE

Additional Editor Comments:

N/A

Reviewers' comments:

Reviewer's Responses to Questions

**Comments to the Author**

1. If the authors have adequately addressed your comments raised in a previous round of review and you feel that this manuscript is now acceptable for publication, you may indicate that here to bypass the “Comments to the Author” section, enter your conflict of interest statement in the “Confidential to Editor” section, and submit your "Accept" recommendation.

Reviewer #1: (No Response)

Reviewer #2: (No Response)

Reviewer #3: (No Response)

2. Is the manuscript technically sound, and do the data support the conclusions?

Reviewer #1: Yes

Reviewer #2: (No Response)

Reviewer #3: Partly

3. Has the statistical analysis been performed appropriately and rigorously? 

Reviewer #1: Yes

Reviewer #2: (No Response)

Reviewer #3: No

4. Have the authors made all data underlying the findings in their manuscript fully available?

Reviewer #1: Yes

Reviewer #2: (No Response)

Reviewer #3: Yes

5. Is the manuscript presented in an intelligible fashion and written in standard English?

Reviewer #1: No

Reviewer #2: (No Response)

Reviewer #3: Yes

6. Review Comments to the Author

Reviewer #1: Most of the previous concerns were addressed in the revised manuscript. However, after further examination, there are still some issues that need to be addressed before bringing this up to a publishable standard, please find them below.

1. Abstract, results, please add more details based on the main findings of this study, for instance, the correlation results.

2. Introduction, this part includes too many paragraphs, please consider merging some and further strengthening the logical structure of this part.

3. “Due to the high rate of recurrence of these lesions by conservative methods, several surgeons around the world have reported their techniques with their advances and limitations.”, thus what are the advances and limitations of these techniques? The novelty and significance of this study should further be highlighted.

4. Materials and methods, “Patient population and study area”, “Patient recruitment and selection”, could be merged into one part. What were the exclusion criteria?

5. Materials and methods, “Surgical procedures and rehabilitation”, it would be much clearer if the authors explained the surgery process step by step.

6. Materials and methods, please add one more paragraph explaining all measured variables.

7. Results, it is suggested that the authors should include some sub-sessions here for a better description. And please try to condense these contents and only present the main findings here.

8. Discussion, this part also includes too many paragraphs, please consider merging some and further strengthening the logical structure of this part. Moreover, it is suggested that the authors should start by reiterating the main research objectives of this paper, briefly explaining the main findings of this study, and presenting deeper information in this part.

9. In general, please further strengthen the logical structure of this study, and go through the whole manuscript to avoid any grammar mistakes.

Reviewer #2: (No Response)

Reviewer #3: I am concerned by Table 1. There are multiple cells with either no responses or very few responses. It is usual to combine cells when the numbers in each cell are below an acceptable level (say 5 in each cell). Without this, the statistics become meaningless

7. PLOS authors have the option to publish the peer review history of their article (what does this mean?). If published, this will include your full peer review and any attached files.

Reviewer #1: No

Reviewer #2: No

Reviewer #3: **Yes: **Peter J Franks

---

## [Author Response · Author response to Decision Letter 1]

8 Feb 2023

RESPONSE TO REVIEWERS

PONE-D-22-04569

EPIDEMIOLOGICAL CLINICAL PROFILE AND CLOSURE OF CHRONIC PLANTAR ULCERS IN PATIENTS WITH LEPROSY SEQUELAE UNDERGOING ORTHOPEDIC SURGERY IN A CITY IN WESTERN AMAZONIA

Reviewer #1: 

1. Abstract, results, please add more details based on the main findings of this study, for instance, the correlation results.

Response: More details were added in the results section based on the main findings of this study and their correlations as described in Page 2, lines 35 to 39

2. Introduction, this part includes too many paragraphs, please consider merging some and further strengthening the logical structure of this part.

Response: A few paragraphs were merged on the Introduction and the research’s logical structure was strengthened furthermore as demonstrated in pages 3 and 4; lines 49 to 99.

3. “Due to the high rate of recurrence of these lesions by conservative methods, several surgeons around the world have reported their techniques with their advances and limitations.”, thus what are the advances and limitations of these techniques? The novelty and significance of this study should further be highlighted.

Response: The advances and limitations are: Advances in the use of grafts and skin flaps for the success of plantar ulcer closure. Limitations on high relapses rates as a result, as described on page 4, line 79 to 81. The innovation and importance of this study were also highlighted as described on page 4, lines 86 to 99.

4. Materials and methods, “Patient population and study area”, “Patient recruitment and selection”, could be merged into one part. What were the exclusion criteria?

Response: They were gathered in one part: "Patient population and study area", and "Recruitment and selection of patients", as demonstrated on page 5 lines 110 to 117. Exclusion criteria were included according to page 5, lines 118 to 120.

5. Materials and methods, “Surgical procedures and rehabilitation”, it would be much clearer if the authors explained the surgery process step by step.

Response: They were included on the section “Surgical Procedure and Rehabilitation”. The step-by-step section of the surgical procedure as described on page 5, lines 127 to 138.

6. Materials and methods, please add one more paragraph explaining all measured variables.

Response: One more paragraph has been added explaining all measured variables as described on page 7, lines 168 to 170. 

7. Results, it is suggested that the authors should include some sub-sessions here for a better description. And please try to condense these contents and only present the main findings here.

Response: We included sub-sessions for better description of the results and also did some merging highlighting the main findings according to pages 7 and 8, lines 175 to 205.

8. Discussion, this part also includes too many paragraphs, please consider merging some and further strengthening the logical structure of this part. Moreover, it is suggested that the authors should start by reiterating the main research objectives of this paper, briefly explaining the main findings of this study, and presenting deeper information in this part.

Response: In this part we merged a few paragraphs as well strengthening furthermore the study’s logical structure. We also reiterate the main objectives of the research, briefly explaining the main findings of the study as found on pages 10 to 12, lines 218 to 283.

9. In general, please further strengthen the logical structure of this study, and go through the whole manuscript to avoid any grammar mistakes.

Response: In general, we reinforced the logical structure and highlighted the main objective of the study. We reviewed the manuscript several times correcting grammatical errors as evidenced in the text in pages 2 to 16, lines 22 to 409.

Reviewer #3: 

I am concerned by Table 1. There are multiple cells with either no responses or very few responses. It is usual to combine cells when the numbers in each cell are below an acceptable level (say 5 in each cell). Without this, the statistics become meaningless

Response: Table 1 was adjusted with the p-value and the demonstration of absolute and relative frequency eliminating the cells without answers or with few answers accordingly to page 9, lines 211 to 213.

---

## [Decision Letter · Decision Letter 2]

15 Feb 2023

PONE-D-22-04569R2Epidemiological clinical profile and closure of chronic plantar ulcers in patients with leprosy sequelae undergoing orthopedic surgery in a municipality in western Amazon.PLOS ONE

Dear Dr. João,

Thank you for submitting your manuscript to PLOS ONE. After careful consideration, we feel that it has merit but does not fully meet PLOS ONE’s publication criteria as it currently stands. Therefore, we invite you to submit a revised version of the manuscript that addresses the points raised during the review process.

We look forward to receiving your revised manuscript.

Kind regards,

Yaodong Gu

Academic Editor

PLOS ONE

Journal Requirements:

Additional Editor Comments (if provided):

Please check the tables as reviewer suggested.

Reviewers' comments:

Reviewer's Responses to Questions

**Comments to the Author**

1. If the authors have adequately addressed your comments raised in a previous round of review and you feel that this manuscript is now acceptable for publication, you may indicate that here to bypass the “Comments to the Author” section, enter your conflict of interest statement in the “Confidential to Editor” section, and submit your "Accept" recommendation.

Reviewer #1: All comments have been addressed

Reviewer #3: (No Response)

2. Is the manuscript technically sound, and do the data support the conclusions?

Reviewer #1: (No Response)

Reviewer #3: Partly

3. Has the statistical analysis been performed appropriately and rigorously? 

Reviewer #1: (No Response)

Reviewer #3: No

4. Have the authors made all data underlying the findings in their manuscript fully available?

Reviewer #1: (No Response)

Reviewer #3: No

5. Is the manuscript presented in an intelligible fashion and written in standard English?

Reviewer #1: (No Response)

Reviewer #3: Yes

6. Review Comments to the Author

Reviewer #1: (No Response)

Reviewer #3: Unfortunately the authors have failed to make any changes to table 1 as I requested, and as such I cannot recommend publication in its present form. To be assured of an appropriate p-value it is recommended that around 5 responses should appear in each cell of the table. I would recommend the following changes

1. Schooling. Incomplete High school, High School and Higher Education should be combined. While this will not give 5 responses in each cell this will give a more robust p-value

2. Marital status. The Missing option should be excluded from the analysis as this is not a legitimate response.

3. Time with ulcer. 31-40, 41-50 and >50 should be combined to give value for >30 and give a reasonable count in all cells.

4. Subscript text should be deleted. There were no confidence intervals generated.

5. The main results text may need alteration to take into account the changes to the table.

7. PLOS authors have the option to publish the peer review history of their article (what does this mean?). If published, this will include your full peer review and any attached files.

Reviewer #1: No

Reviewer #3: **Yes: **Peter J Franks

---

## [Author Response · Author response to Decision Letter 2]

15 Mar 2023

RESPONSE TO REVIEWERS

PONE-D-22-04569

EPIDEMIOLOGICAL CLINICAL PROFILE AND CLOSURE OF CHRONIC PLANTAR ULCERS IN PATIENTS WITH LEPROSY SEQUELAE UNDERGOING ORTHOPEDIC SURGERY IN A CITY IN WESTERN AMAZONIA

Journal Requirements:

1- Please review your reference list to ensure that it is complete and correct. If you have cited papers that have been retracted, please include the rationale for doing so in the manuscript text, or remove these references and replace them with relevant current references. Any changes to the reference list should be mentioned in the rebuttal letter that accompanies your revised manuscript. If you need to cite a retracted article, indicate the article’s retracted status in the References list and also include a citation and full reference for the retraction notice.

Response Updated and relevant references.

Updated and relevant references marked in red are new references and contextualized according to the text data.

1. WHO. Global update on leprosy (leprosy) [Internet]. 2020 [Accessed November 11, 2021]. Available from: https://www.who.int/publications-detail-redirect/who-wer9636-421-444. .(Reference Kept )

2. Brazil. Ministry of Health. Health Surveillance Secretariat. Leprosy in Brazil: Characterization of physical disabilities. Brasília - DF, 2020. .( Reference Kept)

3. Souza CDF, Fernandes TRMO, Matos TS, Ribeiro-Filho JM, Almeida GKA, Lima JCB et al. Degree of physical disability in the elderly population affected by leprosy in the state of Bahia, Brazil. Acta Fisiatr . 2017; 24(1): 27-32. .( Reference Kept)

4. Brazil , MS. Manual for the prevention of disabilities. Brasília - DF, 2008.. ( Reference Kept)

5. Singh M, Pawar M. Efficacy of Topical Insulin Therapy for Chronic Trophic Ulcers in Patients with Leprosy: A Randomized Interventional Pilot Study. Adv Skin Wound Care. 2020 Feb;33(2):1-6.. ( Reference updated )

6. Serrano-Coll H & Cardona-Castro N.Neuropathic ulcers in leprosy: clinical characteristics, diagnosis and treatment. J Wound Care . 2022; 31(Sup6): 32-40.( Reference Kept)

7. Riyaz N, Sehgal VN. Leprosy: trophic skin ulcers. Skin Med 2017;15(1):45–51. .( Reference updated )

8. Gahalaut P, Pinto J, Pai GS, Kamath J, Joshua TV. A new treatment for plantar ulcers in leprosy: local superficial flaps. Lepr Rev. 2005;76(3): 220-31. . ( Reference Kept)

9. Price EW. The problem of the plantar ulcer. Lepr Rev. 1964; 35: 267-72. . ( Reference Kept)

10. Srinivasan H, Mukherjee SM: Trophic ulcers in leprosy III. Surgical treatment of chronic foot ulceration. Leprosy India . 1964; 36: 186-193. . ( Reference Kept)

11. Cohen JC, of Miranda ST . Orthopedic Surgical Treatment of Foot in Leprosy. Orthop Clin North Am. 2020; 51(2): 279-91. . ( Reference Kept)

12. Griffts G. Notes on treatment of ulcers in leprosy patients with Polybactrin. Lep Rev. 1966;37(4):227-9. .( Reference updated )

13. Batista KT, Monteiro GB, Y-Schwartzman UP, Aureliano A, Rosa AG, Correia CZ, et al. Treatment of plantar ulcer stemming from leprosy. Rev. Bras. Plastic Cir. 2019; 34: 497-503. . ( Reference Kept)

14. Rao KS, Mahendra Kumar, Oommen PK, Swamy MK, Selvaganapathy S. Collagen leaf and its usefulness in healing ulcers in leprosy patients. Ind J Lepr . 1987; 59: 435-441. . ( Reference Kept)

15. Hu X, Sun H, Han C, Wang X, Yu W. Topically applied rhGM-CSF for the wound healing: a systematic review. Burns. 2011;37(5):729-41. .( Reference updated )

16. Noe JM, Barber J. Chronic leg ulceration in a patient with leprosy. West J Med. 1974;121(5):430-32. .( Reference updated )

17. Cruz H, Kulkarni VN, Dey A, Rendall G. Plantar ulceration in leprosy patients. J Wound Care . 1996; 5(9): 406-11. . ( Reference Kept)

18. Lopes GDF . Efficacy and efficiency of insoles in the prevention and rehabilitation of plantar ulcers in neuropatic feet in leprosy and diabetes [dissertation]. Bauru (SP): Lauro de Souza Lima Institute; 2022. . ( Reference Kept)

19. Noordeen SK, Srinivasan H. Deformity in leprosy: an epidemiological study. Indian J Med Res. 1969; 57(1): 175-81. . ( Reference Kept)

20. GUIMARÃES, Heloísa Cristina Quatrini Carvalho Passos et al. Scientific evidence on leg ulcers as leprosy sequel. Acta Paulista de Enfermagem, v. 32, p. 564-570, 2019. .( Reference updated )

21. Jin Y, Tan Y, Wang J, Zhong H, Yue J, Li H and Yan L. Effect of different surgical methods on leprosy plantar ulcers. Chinese journal of Repairer and Reconstructive Surgery. 2009; 23 (10): 1183-1186. . ( Reference Kept)

22. Reis FJJD, Gomes MK, Cunha AJLAD. Evaluation of the limitation of daily activities and quality of life of leprosy patients undergoing neurolysis surgery for neuritis treatment. Physiotherapy and Research. 2013; 20: 184-190. . ( Reference Kept)

23. Bernardin R, Thomas, B. Surgery for neuritis in leprosy: indications and results of different types of procedures. Lepr Rev. 1997; 68(2): 147-154. . ( Reference Kept)

24. Hidalgo DA, Shaw WW. Anatomical bases of plantar flap design. plastic Reconstr Surg. 1986; 78(5): 627-36. . ( Reference Kept)

25. Yan L, Zhang G, Zheng Z, Li W, Zheng T, Watson JM, Piefer A. Comprehensive treatment of complicated plantar ulcers in leprosy. Chin Med J ( Engl ). 2003; 116(12): 1946-8. ( Reference Kept)

26. Shukla VK, Tiwary SK, Barnwal S, Gulati AK, Pandey SS. Effect of autologous graft cell suspension on chronic wounds that do not heal: a pilot study. Can J Surg. 2010; 53: 6-10. . ( Reference Kept)

27. Nagaraju U, Kashyap P, Raveendra L. Autologous smashed follicular dermal graft with epidermal cell suspension in chronic nonhealing leg ulcers. J Cutan Aesthet Surg. 2020; 13(1): 38-42. . ( Reference Kept)

28. Dheemant M, Yadalla HK, Raju BP. Efficacy of autologous crushed follicular dermal graft and epidermal cell suspension in the treatment of chronic trophic ulcers that do not heal in leprosy patients. Indian Dermatol Online J. 2021; 12(6): 868-872. . ( Reference Kept)

29. Dong L, Li F, Wang Z, Jiang J, Zhang G, Peng J, et al. Techniques for covering soft tissue defects resulting from plantar ulcers in leprosy: Part General considerations and summary of results. Indian J Lepr.1999; 71(3): 285-95. . ( Reference Kept)

30. Santana EMF, Antas EMV, Brito KKG & Silva MA. Profile of leprosy patients in a secondary care center. UFPE Nursing Journal. 2017; 11(11): 4404-9. . ( Reference Kept)

31. Lustosa AA, Nogueira LT, Pedrosa JIS, Teles JBM & Campelo V. The impact of leprosy on health-related quality of life. Rev Soc Bras Med Trop. 2011; 44(5): 621-6. . ( Reference Kept)

32. Upputuri B, Srikantam A, Mamidi RS. Comorbidities associated with non-healing of plantar ulcers in leprosy patients. PLoS Negl Trop Dis. 2020; 14(6): . ( Reference Kept)

33. Pinheiro CIP, Moreira ICCC, Nunez SC, Silva TB, Pereira MS, Campelo DP et al. Clinical-epidemiological profile of people affected by neurotrophic ulcers resulting from leprosy. Res Soc and Develop. 2021; 10(12): . ( Reference Kept)

34. Mustapha G, Obasanya OJ, Adesigbe C, Joseph K, Nkemdilim C, Kabir M, et al. Occurrence of plantar ulcer among leprosy patients in northern Nigeria: a study of contributing factors. Ann Afr Med. Marchar. 2019; 18(1): 7-11. . ( Reference Kept)

35. Ebenso J, Muyiwa LT, Bassey E. Self-care and ulcer prevention groups in Okegbala, Nigeria. Lepr Rev. 2009; 80(2): 187-96. . ( Reference Kept)

36. Cheung JTM, Zhang M. Parametric design of pressure relief foot orthosis using the finite calculation method based on statistics. Physics Eng. Med . 2008; 30(3): 267-77. . ( Reference Kept)

37. Veen NHJ, Mcnamee P. Richards JH, Smith WCS. Cost-effectiveness of interventions to prevent disability in leprosy: a systematic review. plos A . 2009; 4(2): . ( Reference Kept)

38. Tashiro S, Gotou N, Oku Y, Sugano T, Nakamura T, Suzuki H, et al. Relationship between plantar pressure and sensory disorders in leprosy patients - Preliminary research and literature review. Sensors (Basel). 2020; 20(23): 6976. . ( Reference Kept)

 References Removed from last version - These references below 09 references have been removed to highlight the above references that are more relevant.

1. Talhari C, Talhari S, Pena GO. Clinical Aspects of Leprosy. Dermatol CLin. 2015; 33: 26-37.

2. Haythornthwaite HW: Closed plaster for trophic ulcers. Leprosy India . 1943; 15: 20-22.

3. Rao KS, Mahendra Kumar, Oommen PK, Swamy MK, Selvaganapathy S. Collagen leaf and its usefulness in healing ulcers in leprosy patients. Ind J Lepr . 1987; 59: 435-441.

4. Pieri FM,Ramos ACV, Crispim JA, Pitiá ACA,Rodrigues LBB, Silveira TRS, et al. Factors associated with disabilities in patients diagnosed with leprosy:a cross-sectional study. hansenology Internacional,2012; 37(12): 22-30.

5. Budel AR, Raymundo AR, Costa CF, Gerhardt C, Pedri LE. Profile of leprosy patients treated at the dermatology outpatient clinic of the Evangelical Hospital of Curitiba. A Bras Dermatol. 2011; 86: 942-6.

6. Batista ES, Campos RX, Queiroz RCG, Siqueira, S. L., Pereira, S. M., Pacheco, T. J., Pessanha, T. O. et al. Socio-demographic and clinicalepidemiological profile of patients diagnosed with leprosy in Campos dos Goytacazes, RJ*. Revista Brasileira de Clinica Médica. 2011; 9(2): 101-6.

7. Miranzi SSC; Pereira LHM & Nunes AA. Epidemiological profile of leprosy in a Brazilian municipality in the period from 2000 to 2006. Rev Soc Bras Med Trop. 2010; 43(1): 62-67.

8. Barreto JG, Bisanzio D, Frade MAC, Moraes TMP, Gobbo AR, Guimarães LS et al. Spatial epidemiology and serological cohorts increase early detection of leprosy. BMC Inf Dis. 2015; 15(1): 1-9.

9. De Oliveira MP, de Sousa JR, de Araujo RS, de Sousa Aarão TL, Quaresma JAS. Protein profile of leprosy patients with plantar ulcer scans from the Eastern Amazon region. Infect dis Poverty . 2017; 6(1): 105.

2- Additional Editor Comments (if provided):

Please check the tables as reviewer suggested.

Response: The table 1 has been revised and it is below

3- Reviewer #3:

 Unfortunately, the authors have failed to make any changes to table 1 as I requested, and as such I cannot recommend publication in its present form. To be assured of an appropriate p-value it is recommended that around 5 responses should appear in each cell of the table. I would recommend the following changes.

1. Schooling. Incomplete High school, High School and Higher Education should be combined. While this will not give 5 responses in each cell this will give a more robust p-value 

Response: schooling : incomplete high school, high school and Higher education were joined and added together according to table 1, taking into account the Reviewer's guidelines

2. Marital status. The Missing option should be excluded from the analysis as this is not a legitimate response.

Response: Marital status: The option “ Missing was removed from table 1.

3. Time with ulcer. 31-40, 41-50 and >50 should be combined to give value for >30 and give a reasonable count in all cells.

Response: Time with ulcer. 32-40 , 41-50 and >50 the data was combined and gave a value of >30

4. Subscript text should be deleted. There were no confidence intervals generated.

Response: Subscript text from table 1 was removed.

5. The main results text may need alteration to take into account the changes to the table.

Response: Text changed: 

 Summary, Results and Discussions the main results taking into consideration the changes made in table 1 as directed by the Reviewer. Changes observed in the final text.

---

## [Decision Letter · Decision Letter 3]

6 Apr 2023

Epidemiological clinical profile and closure of chronic plantar ulcers in patients with leprosy sequelae undergoing orthopedic surgery in a municipality in western Amazon.

PONE-D-22-04569R3

Dear Dr. João,

We’re pleased to inform you that your manuscript has been judged scientifically suitable for publication and will be formally accepted for publication once it meets all outstanding technical requirements.

Kind regards,

Yaodong Gu

Academic Editor

PLOS ONE

Reviewers' comments:

Reviewer's Responses to Questions

**Comments to the Author**

1. If the authors have adequately addressed your comments raised in a previous round of review and you feel that this manuscript is now acceptable for publication, you may indicate that here to bypass the “Comments to the Author” section, enter your conflict of interest statement in the “Confidential to Editor” section, and submit your "Accept" recommendation.

Reviewer #3: All comments have been addressed

2. Is the manuscript technically sound, and do the data support the conclusions?

Reviewer #3: Yes

3. Has the statistical analysis been performed appropriately and rigorously? 

Reviewer #3: Yes

4. Have the authors made all data underlying the findings in their manuscript fully available?

Reviewer #3: Yes

5. Is the manuscript presented in an intelligible fashion and written in standard English?

Reviewer #3: Yes

6. Review Comments to the Author

Reviewer #3: This is the third iteration of this paper. My previous comments were on the statistics used in the table, These have now been addressed.

7. PLOS authors have the option to publish the peer review history of their article (what does this mean?). If published, this will include your full peer review and any attached files.

Reviewer #3: **Yes: **Peter J Franks

---

## [Editor Report · Acceptance letter]

13 Apr 2023

PONE-D-22-04569R3 

Epidemiological clinical profile and closure of chronic plantar ulcers in patients with leprosy sequelae undergoing orthopedic surgery in a municipality in western Amazon. 

Dear Dr. João:

I'm pleased to inform you that your manuscript has been deemed suitable for publication in PLOS ONE. Congratulations! Your manuscript is now with our production department. 

Kind regards, 

on behalf of

Professor Yaodong Gu 

Academic Editor

PLOS ONE